# Reading Behind Bars: Literacy and Survival in U.S. Prison Literature

**Katie Owens-Murphy**

Department of English, University of North Alabama, Florence, AL 35632, USA; kowensmurphy@una.edu

**Abstract:** This paper unpacks the contradiction between the benefits of literacy and the punitive prison policies that seek to curb or regulate reading by beginning with the complicated historical relationship between incarceration and literacy. I then turn to the testimonies of two prominent incarcerated autodidacts who I now regularly teach within my prison literature classes both on my university campus and at a men's prison. The writings of Malcolm X and Etheridge Knight model the difficulties of negotiating the institutional risks and personal and political rewards of learning to read and write behind bars—particularly while Black. What is more, while literacy may provide an "on-ramp" toward higher education, barriers for incarcerated people continue to proliferate in our current era in the form of book bans, paywalls, and the material conditions of prisons themselves.

**Keywords:** prisons; literacy; reading in prison; book bans

## 1. Introduction

I began teaching literature at prisons and jails as a classroom aide in the fall of 2011 while finishing my doctoral degree in central Pennsylvania, which was home to three—now four—large maximum-security state prisons within a 60-mile radius. Tasked with preparing incarcerated students to earn their GED diplomas through a study of William Golding's *Lord of the Flies*, a text assigned by the lead teacher, I struggled to engage adult students who had long since left or been dismissed from school; so when I came across a provocative article by H. Bruce Franklin entitled "Can the Penitentiary Teach the Academy How to Read?" (Franklin 2008) that argued the merits of Donald Goines' street lit, I swapped out Golding for Goines and his predecessor, Iceberg Slim, in our small weekly reading groups. It was a game-changer. Our classroom set of books disappeared and circulated among the cellblocks. Others who were not in our reading group pleaded for spare copies. Everyone was reading voraciously. I was proud of our program. A week later, we were given a decree from the warden: no more street lit, and students were now only allowed one book in their cells at a time—a rule that openly interfered with our homework assignments.

Book bans are common practice within correctional facilities: as Megan Sweeney (2008) has argued, prisons use reading restrictions to flaunt their power, attempting "to control prisoners by controlling their reading" (p. 667). To what exactly the warden objected remains a mystery because he never felt pressed to justify his decision even to the facility's teachers. Yet these bans run counter to the purportedly rehabilitative aims of the Department of Corrections, undermining the prominent and enduring tradition of literacy as a means of self-transformation that includes the famous example of Malcolm X. In fact, Malcolm X's *Autobiography* marks the emergence of an important trope within the canon of contemporary U.S. prison writing that positions literacy in prison as a matter of survival in literal as well as figurative senses. This sentiment is echoed in the recent *New York Times* Op Ed by Christopher Blackwell (2022), "Reading While Incarcerated Saved Me. So Why Are Prisons Banning Books?"

In what follows, I unpack the punitive prison policies and environments that seek to curb or regulate reading by beginning with the complicated historical relationship between

incarceration and literacy. I then turn to the testimonies of two prominent incarcerated autodidacts whose works I now teach regularly on campus as well as at a men's prison. The writings of Malcolm X and Etheridge Knight model the difficulties of negotiating the institutional risks and personal and political rewards of learning to read and write behind bars—particularly while Black. They also exemplify very different survival strategies in response to the collision between literacy and structural racism. While Malcolm X turned outward, using his newfound pulpit to proselytize, Knight turned inward, opting instead for lyric poetry to render his experiences visible. Having eulogized Malcolm X in his first published poem, Knight was well aware that combining literacy with the politics of Black power—both inside and outside of prison—made Black men vulnerable to incredible violence.

Still, both writers understood that literacy also holds practical value for people ensnared in the criminal-legal system, many of whom must navigate their own cases and appeals. Indeed, this is one of the reasons why carceral facilities are required to maintain law libraries. Malcolm X's prison term (1946–1952) marked the beginning of the era of the Warren Court that came to be known as the "prisoners' rights movement" during which incarcerated people successfully lobbied for the right to file a writ of habeas corpus (*Jones v. Cunningham*, 1963) as well as the right to sue for civil rights violations in federal court (*Cooper v. Pate*, 1964). But by 1971, mounting frustrations culminating in the Attica Prison Uprising proved how difficult these rights were to realize in practice. In fact, the striking similarities that Malcolm X's narrative shares with later accounts by Etheridge Knight and by contemporary autodidacts reveal how little the overall climate of corrections has changed over time, especially with regard to reading access, which remains precarious at best.

Life writing in prison invites us to look beyond overworked binaries that "all too often represent life writing as a liberatory practice and the prison as an oppressive and restrictive institution," as Simon Rolston writes (Rolston 2021, p. 4). He continues,

> To navigate the role of the prison in their lives and work, incarcerated people make surprising innovations that redefine what can and what cannot be said in autobiographical discourse; in this way, they challenge how we think about life writing and institutional power. (p. 5)

The writings of Malcolm X and Etheridge Knight—particularly when read together—reveal the complex negotiations that Black autodidacts must make as they navigate literacy in carceral contexts. The expressions and admonitions they yield are especially important for those of us who work in higher education in prison, where incarcerated students continue to live with these "complicated tensions of power and resistance" in our classrooms (p. 6).

## 2. Reading the Modern American Prison

The prison conditions outlined in the works of Malcolm X and Etheridge Knight dramatize the conflict between rehabilitation and punishment that date back to the construction of its first dedicated carceral space in the U.S, Philadelphia's Walnut Street Jail (1773). The jail was created as a place for incapacitation that juxtaposed itself to European models of capital and corporal punishment based on retribution. An early attempt at penal reform, the Walnut Street Jail—which quickly became overcrowded—added an experimental "penitentiary house" in 1790 that featured sixteen single cells designed for solitary confinement.

Though we now have a more robust understanding of the psychological harm solitary confinement produces,[1] isolation and religious reflection became the model for Eastern State Penitentiary (ESP), the first dedicated prison in the U.S. Completed in 1829, the prison contained a panoptical radial floor plan containing single-occupancy cells that each featured a food hatch, a one-way peephole, and a skylight that simulated the eye of God and the light of heaven. The ostensible focus on penitence, reform, and redemption became the hallmark of the Pennsylvania system, which was based on socially progressive Enlightenment ideals as well as "the Quaker belief that any man—even a criminal—possessed a share of the

divine 'inner light' which could be reached and nurtured through proper treatment" (Lewis 1965, p. 3). The isolation model, however, soon proved too costly to sustain, as each eight-by-ten foot cell contained its own central heating and plumbing. ESP began to recoup some of these costs by incorporating labor into its model, encouraging solitary "handicrafts" such shoemaking, chair-making, and hand loomed weaving (pp. 164–65).

Still, ESP could not remain competitive in the wake of another carceral model that explicitly abandoned reflection for manufacturing. Auburn State Prison (1818), which became a synecdoche for the "New York" or "congregate" system, featured communal workshops in which residents labored together by day; by night, they returned to solitary cells much smaller in their dimensions than those at ESP. Auburn simulated ESP's isolation by requiring residents to labor in total silence, a condition that was reinforced through strict corporal punishment. Auburn Prison resident Austin Reed—author of the earliest-known African American prison memoir—reports conditions that approach slavery as he sustains a whipping from "the Keeper" for laughing out loud: "He drew out a blue raw hide . . . he then gave me seven cuts on the back and told me . . . the next act he caught me in, he would put something else on my back which would make me buldge [sic]" (Reed 2016, p. 150). Unlike ESP, Auburn also opted for a labor system that ceded control and financial risk to private industry in the interests of profit. Auburn's very floor plan reveals its profound investment in manufacturing: as of 1828, the prison contained a blacksmith shop, dye house, weavers' shop, tailors' shop, shoe shop, tool room, and cooper's shop with extra room for "new shops to be built" (Lewis 1965, Illustration 4). The congregate model proved profitable, especially in the hands of one of its principal engineers, notorious prison warden Elam Lynds, who ruled through a brutal combination of intimidation and corporal punishment with the help of one of his favorite tools, the cat'o'nine tails (Conover 2000, p. 180). Lynds helped to yield impressive profits both for the prison and for private industry. Beginning in 1830, Auburn began to earn "small annual surpluses aggregating slightly over $29,000"—a value of over $938,000 today (Lewis 1965, p.186). The fusion of retribution and capitalism within the modern penitentiary was now complete: by 1913, the Pennsylvania System was "officially discontinued" in the U.S.

Though they were in direct competition with one another, these early carceral models share an important swatch of common ground, as Jodi Schorb (2014) has argued: the state's unwillingness to invest in literacy. Indeed, the only reading material allotted to residents under both systems was a copy of the Bible. "Literacy remained a minor refrain in reformist discourse and the public debates about the purposes, best practices, and social good that might come of the prison," writes Schorb (p. 3). Though both systems were early to recognize illiteracy as a major challenge for their residents, neither made any accommodations for literacy training until the mid-nineteenth century.[2] Few reformers believed in the transformative power of literature, and fewer still believed that prisoners possessed "the capacity and discipline for self-corrective reading" (p. 121).

Unfortunately, the legacy of the Pennsylvania/congregate rivalry is that the U.S. inherited the worst elements of both systems, including their shared pessimism toward the value of reading. Today, educational programs subsidized by state funding are largely limited to GED exam preparation and trade school certification, and reading remains subsidiary to the commodification of labor whereby private or state-owned "correctional industries" profit by paying workers an average of 33 cents to $1.41 per hour to create office furniture and textiles (Sawyer 2017). What is more, the post-1950s era of "corrections" that expanded and codified prison administrations have strengthened the chain of command through which wardens such as Lynds may exercise complete control over residents in ways that even diminish the authority of the U.S. Constitution.[3] A 9-0 decision in *Overton v. Bazzetta*, for example, found that "the burden . . . is not on the state to prove the validity of prison regulations but on the prisoner to disprove it" (Kennedy 2003), and a 6-2 decision in *Beard v. Banks* (Breyer 2006) ruled in favor of the Pennsylvania Department of Corrections' policy to withhold reading material from people incarcerated in the Long Term Segregation Unit.

The historical challenges of overcrowding, physical and psychological violence, and authoritative control and exploitation course through the literacy narratives of incarcerated writers who struggle to read against the severe limitations these conditions imposed on their intellectual lives. Both Malcolm X and Etheridge Knight credit literacy with their survival within a system that continuously tried to erase their humanity through heightened surveillance, isolation, and bodily harm. Both writers also attribute these risks and punishments to their positionality as Black men. Placed in these historical and material contexts, the literature of Malcolm X and Knight lays important groundwork for our understanding the intersection of race and gender within the criminal-legal system and the material conditions of imprisonment that render literacy both so important and so precarious.

### 3. "Behind Bars, a Man Never Reforms": Malcolm X at Charlestown State Prison; Massachusetts Reformatory at Concord; and Norfolk Prison Colony, 1946–1953

Malcolm X paradoxically draws on the rhetoric from his Baptist upbringing to articulate his conversion to the Nation of Islam in "Saved," the chapter that serves as one of the crucial turning points in his autobiography. His conceit of one who is "born again" is a powerful symbol for the new life that literacy offers him during the seven years he serves in Massachusetts correctional facilities for burglary. As Rolston (2021) points out, life writing in prison often hinges on the trope of "conversion" that maps onto histories of penology, autobiography, and literacy, even making use of all three discourses, as is the case in the *Autobiography of Malcolm X* (Rolston 2021, pp. 10–12). Critics who discuss this period of Malcolm X's life focus exclusively on his experience at Norfolk Prison Colony, an experimental institution noted for its unique focus on education and which provided the setting for his dramatic turnaround. Relatively little attention has been paid to Malcolm's term at Charlestown State Prison, where he began—and ended—his lengthy sentence, and where he first discovered the possibilities of literacy. The conditions he endured at Charlestown not only point to the wide variation among prison facilities within a single state system, they underscore the sheer improbability of learning to read behind bars in facilities that lack basic resources for wellness and hygiene.

Malcolm X's literacy narrative is often said to begin with incarceration, but the reverse is also true: his incarceration narrative begins with illiteracy. "Shorty didn't know what the word 'concurrently' meant," declares the sentence that famously opens the chapter "Satan" (Malcolm and Haley 1965a, p. 154). As the judge issues sentencing for fourteen felony counts to be "run concurrently" for Malcolm X and his co-defendant, Shorty panics: "Shorty, sweating so hard that his black face looked as though it had been greased, and not understanding the word 'concurrently,' had counted in his head to probably over a hundred years; he cried out, he began slumping. The bailiffs had to catch and support him." The incident with Shorty illustrates the high stakes of illiteracy in the criminal-legal system: Shorty can't understand the language that is used during his own sentencing.

"Satan" and its companion chapter, "Saved," are frequently excerpted as evidence of literacy's potential to lend confidence and authority to developing readers. They chronicle Malcolm Little's transformation from Harlem Hustler to political activist during his lengthy prison term for armed robbery. Yet this chapter also showcases the very real ramifications of illiteracy for Malcolm X and Shorty. With little formal education or legal acumen, they struggle to understand why their lawyers and case workers are more concerned with their co-defendants, two white women, than with the commission of the crime itself. "How, where, when, had I met them? Did we sleep together? Nobody wanted to know anything at all about the robberies," Malcolm X recounts (p. 153). Sentencing disparities reflecting race and gender are also prominently displayed here: though Malcolm "got ten years," the women are sentenced to "one to five" (p. 154).[4] In retrospect, Malcolm the narrator notes that the "average burglary sentence for a first offender, as we all were, was two years"—a statistic that was unavailable to him at the time of his sentencing (p. 153). Literacy became instrumental in helping Malcolm to understand his prison term as part of a larger pattern

of white supremacy in the U.S., one that had left indelible imprints all over his life, from his father's untimely death (likely the work of the Black Legion) to his English teacher's suggestion that he consider carpentry rather than law, which was "no realistic goal for a n——" (p. 38)—a comment that, by Malcolm X's account, effectively drove him from academics to the streets and, ultimately, to Charlestown State Prison by the age of twenty.

Charlestown was an unlikely place for Malcolm X to recommit himself to education. Built in 1805, condemned in 1878, and reopened six years later to alleviate overcrowding from MCI-Concord, Charlestown had scarcely modernized by the time Malcolm X entered there in 1946. It did not contain any plumbing or running water; instead, residents used covered pails for toilets that were to be emptied once every 24 h. Malcolm X impresses upon his readers the mental distress that results from such unsanitary conditions. "I don't care how strong you are," he writes. "You can't stand having to smell a whole cell row of defecation" (p. 155). His study space was a "dirty, cramped cell" in which he could lie down and "touch both walls." Outside of his cell, Malcolm labored in a congregate-style factory where he painted license plates on a conveyor belt without any financial compensation, since Massachusetts was one of twenty-five states that denied wages to incarcerated workers (Gambino 1993, p. 19). His leisure time was also compromised by those distractions that flourish behind prison walls—the stolen nutmeg, for example, on which Malcolm X says he and his cellmate got high in order to forget the squalor of their surroundings (Malcolm and Haley 1965a, p. 156). At the insistence of another autodidact (John Elton Bembry, called "Bimbi"), Malcolm took advantage of the available college correspondence courses, which included a small selection of offerings rooted in Classical and European languages such as introductory Latin and German (Tucker 2017, p. 199). The availability of these courses, which provided a gateway for Malcolm into the realms of reading, writing, and oratory, is itself a testament to the impact of access to higher education in prison. One year and two courses later, Malcolm X was finally able to "write a decent and legible letter" (Malcolm and Haley 1965a, p. 158), though he still described his reading comprehension as poor. "Every book I picked up had few sentences which didn't contain anywhere from one to nearly all of the words that might as well have been Chinese," he recollects. "When I just skipped those words, of course, I really ended up with little idea of what the book said. . . . [I was] still going through only book-reading motions" (p. 175).

Had Malcolm finished out his sentence in Charlestown State Prison, he may have continued on that way.[5] Fortunately, his sister Ella successfully petitioned for Malcolm's transfer to the Norfolk Prison Colony, established in 1927 by Harvard penologist Howard Belding Gill in the reform-based spirit of the Pennsylvania System. Norfolk became locally famous for its progressive programming, including a debating club that helped Malcolm to sharpen those rhetorical and argumentative strategies he later deployed in his public speeches (Schwartzapfel 2011). Especially noteworthy is the prison's design and architecture, which replaced vertical bars with doors and windows. Reflecting on the conditions at Charlestown, Malcolm X articulates the psychological trauma produced by cages: "Any person who claims to have deep feeling for other human beings should think a long, long time before he votes to have other men kept behind bars—caged," he writes. "I am not saying there shouldn't be prisons, but there shouldn't be bars. Behind bars, a man never reforms" (p. 155). Indeed, Norfolk was unusual in providing material conditions that encouraged incarcerated people to feel and behave more like students. It provided educational programming staffed by instructors from prestigious nearby institutions such as Harvard and Boston University.[6] Most importantly, it had an impressive and accessible library that was unusual in its size and scope because it had received a large donation from a former senator and was deliberately networked with other public libraries in the state (Tucker 2017, p. 196). "At Norfolk," Malcolm X wrote, "we could actually go into the library, with permission—walk up and down the shelves, pick books. There were hundreds of old volumes, some of them probably quite rare. I read aimlessly, until I learned to read selectively, with a purpose" (Malcolm and Haley 1965a, p. 161).

Malcolm X's "purpose," of course, was his growing interest in the teachings of Elijah Muhammad, the charismatic leader of the Nation of Islam whose philosophies of Black empowerment were already appealing to Malcolm's siblings back in Detroit and Chicago. His sister Hilda encouraged Malcolm to write to Muhammad, who "understood what it was to be in the white man's prison," having served five years in a federal prison for dodging the draft (p. 167). Malcolm X's attempts at letter-writing, however, force him to reckon once again with his own frustrations with literacy. "At least twenty-five times I must have written that first one-page letter to him, over and over," he recalls. "I was trying to make it both legible and understandable. I practically couldn't read my handwriting myself; it shames me even to remember it" (p. 172). The words "shame" and "humiliation" pervade this section of the autobiography; Malcolm X concludes his letter to Muhammad by apologizing for his poor penmanship. Yet he is also becoming aware of the power of language to give shape, authority, and understanding to the profound thoughts and feelings that are welling up inside of him but that he is "pitifully unable to express" (p. 173). Here, Malcolm X turns to the metaphor of religious supplication and contemplation. After bending his knees for the first time in prayer, he makes the decision to devote his remaining years of his sentence to become "the nearest thing to a hermit at Norfolk Prison Colony" in order to learn to read and write.[7]

This spiritual-intellectual awakening coincides with Malcolm X's literary awakening, which famously begins with the most prosaic of texts. Overwhelmed by the sheer volume of the dictionary—"I'd never realized so many words existed!"—Malcolm X begins to copy words and definitions "just to start some kind of action" (p. 175). Slowly, copying gives way to other modes of cognition such as memorization, retention, learning, and, finally, understanding. Malcolm's mastery of language is the watershed moment in his autobiography, and it is articulated through the language of adventure and discovery. "Anyone who has read a great deal can imagine the new world that opened," he writes (p. 176). Though Norfolk afforded plenty of opportunities for self- and guided study, he continued to read voraciously in fifty-eight minute intervals after "lights out," squinting to read by the faint glow of the lighted corridor and leaping back into bed when the officers made their hourly rounds. (Here he developed the astigmatism that would require him to wear those iconic glasses.) As Laura Dubek observes, Malcolm begins to cultivate the practice of "cultural literacy"—his ability to read and understand people, signs, and institutions that paralleled his reading and understanding of literature (Dubek 2014, p. 207). Brazilian pedagogue and philosopher Paulo Freire famously articulates this relationship "between *reading words and reading the world*" in depicting his own journey into literacy: "the process of reading reality in which we are enveloped demands, undoubtedly, a certain theoretical understanding of what is *happening* in reality," he asserts (Freire and Horton 1990, p. 31). Literacy, for Freire—and for Malcolm—requires two contextual moves: situating the text within the author's time and place, and situating the text in relation to one's own.

However, Malcolm X's cultural literacy, his way of "reading the world" in his own time and place, also began to attract the attention of the authorities at Norfolk. "The white man is the devil," he readily informed anyone who seemed "ripe" for Muhammad's message (Malcolm and Haley 1965a, p. 186). His conversion to Islam attracted the attention of the local paper, which reported that he was "under close surveillance" by officers (Gambino 1993, p. 14). His letters to and from prison—so crucial to the development of his literacy—also passed through the censor's office. Eventually, the authorities at Norfolk became agitated by his Black power politics and, though they never explicitly condemned Malcolm's behavior, they arranged for his transfer back to Charlestown. "They used as a reason for my transfer that I refused to take some kind of shots, an inoculation or something," he recalls (p. 193). This confrontation, too, was hastened by the material conditions of the prison: according to Perry (1991), "Norfolk's well water was chemically untreated and susceptible to contamination by sewage," so the prison required each resident to receive a typhoid inoculation. Malcolm X refused, since Nation of Islam prohibited the use of "harmful substances" such as drugs and alcohol (p. 132). Thus, he finished the final

year of his sentence once again without any indoor plumbing in a place where he had substantially "less maneuverability" and the only class offered was a Bible study (Malcolm and Haley 1965a, p. 193).

The administration's relocation of Malcolm from Charlestown to Norfolk and back to Charlestown again highlights a number of realities that shape the practice of reading behind bars.[8] Perhaps most importantly, it dramatizes the variability of prison conditions, which differ radically from one facility to another in ways that reflect the tensions between reform and retribution within U.S. penal history. Often, as with Charlestown, these conditions are a function of the prison's age, but they are also a function of administrative leadership. Malcolm X was fortunate to have been transferred to the Norfolk Prison Colony, where wardens Maurice N. Winslow and his successor John J. O'Brien continued to support the prison's mission to loosen punitive restrictions and offer opportunities for education (Bernstein 2010, p. 13; Bosworth 2005, p. 641). Norfolk is often read as the beginning of Malcolm's arc toward self-liberation.

Yet even prisons oriented around rehabilitation prioritize surveillance and control over reform, particularly when faced with the threat of political radicalization that Malcolm X symbolized. In this context, Norfolk's "reform" measures seem specious and superficial: while it may have eliminated vertical bars in favor of a more open floor plan, the prison itself was built by the labor of incarcerated people in 1927 in a form of exploitation inherited from the Auburn model (Sweeney 2013). By the time Malcolm X entered in 1946, Norfolk was already dismantling several of its key reform measures, including a measure that enabled residents to wear plain clothes (Bosworth 2005, p. 641). Most importantly, Malcolm's certainty that his letters were being screened and documented by prison censors who never revealed themselves prior to his transfer reminds us that book bans are not the only tool carceral institutions have for controlling literacy when Black autodidacts become empowered.

In fact, Malcolm X's sudden transfer aligns with the experiences of other contemporary African American writers. In 2012, incarcerated activist Mumia Abu-Jamal was transferred from death row to solitary confinement under "administrative custody" at SCI-Mahanoy under "conditions that are far more restrictive than death row at SCI-Greene" (Tempey 2012). Following extensive media coverage and protests, he was transferred to general population after the state announced that it would no longer pursue his execution. His writings and radio broadcasts, however, were then silenced by a gag order under Pennsylvania's "Revictimization Relief Act," a 2014 law—overturned six months later by a federal judge—that authorized the censorship of prisoners' free speech if it causes "mental anguish" to the victim.[9] More recently, Kevin Rashid Johnson (2018) describes a similar process of censorship, punishment, and isolation in a piece that he managed to release to *The Guardian* in the wake of a nationwide prison labor strike. Johnson—who is not facing a death sentence—was placed in isolation on Virginia's death row because of his persistent advocacy efforts to organize strikes. "That's been the pattern of my incarceration for the past many years," he writes. "I resist, they retaliate." Indeed, Johnson has been transferred to four other states—Texas, Florida, Indiana, and Ohio—since his article's publication before being transferred back to Virginia, where he currently resides.

Malcolm X continued to use incarceration as a metaphor for the structural oppression of Black Americans long after his release. "Don't be shocked when I say I was in prison," he told a 1963 audience. "You're still in prison. That's what American means—prison" (Malcolm and Haley 1965b, p. 8). The relationship between prison and literacy in Malcolm's writings—his self-curated transformation from Harlem hustler to Malcolm X—therefore requires careful contextualization. The tropes of survival that anchor stories such as Malcolm X's are crucial for appreciating the extreme difficulties under which incarcerated writers labor—and labor they do, against all odds. "It is hard/ To make a poem in prison," writes fellow autodidact Etheridge Knight (1968) with characteristic folk simplicity. "The air lends itself not/ to the singer."

### 4. "Soft Songs, like Birds, Die in Poison Air": Etheridge Knight at Indiana State Penitentiary, 1960–1968

Unlike Malcolm X, Etheridge Knight never penned a full-length memoir that details his prison experience, but the following two-line autobiography appeared on the back of each of his chapbooks: "I died in Korea from a shrapnel wound, and narcotics resurrected me. I died in 1960 from a prison sentence and poetry brought me back to life." Whereas Malcolm X articulates his literacy journey through the spiritual currency of conversion, Knight tends to articulate his through the physical body. For Knight, as for Malcolm X, literacy and empowerment are integral to survival in carceral contexts, but they can also threaten survival, as well, by making one more vulnerable to interpersonal and state violence. Knight's poetry employs both liberatory and cautionary rhetoric for Black autodidacts who follow in Malcolm X's footsteps.

Having dropped out of school at age fourteen, Knight was wounded during his service in the Korean War and subsequently became addicted to painkillers. He was convicted in 1960 for an armed robbery in which he snatched an elderly woman's purse in order to fuel his habit. Like Malcolm X, he came to understand the racism that motivated his ten-to-twenty-five-year indeterminate sentence (of which he would ultimately serve eight). "When the detective told me, 'N——, you're goin' to prison!' I believed him," he later told Art Powers (1971, p. 95). Knight became bitter watching men convicted of more serious crimes receive lighter sentences. By the time Art Powers met him, he "didn't have much going for him in confinement. His work record was spotty; he had a limited education . . . [and] wallowed in a haze of red anger and self-pity" (pp. 95–96). In and out of solitary confinement for refusing work duty, Knight was labeled "incorrigible" and transferred from the state reformatory to the Indiana State Prison "with the recommendation that he 'serve his maximum sentence'" (p. 96).

Like Malcolm X before him, Knight turned to reading as a salve for his frustration and as a site for self-discovery: "He read books like they were going out of style and applied himself in many areas—philosophy, art, science, and religion," Powers recounts. "In five years he covered a wide field, and he found a bit of Etheridge Knight in all of them. He found a sense of worth, a yardstick of measurement for himself" (p. 96). Knight began writing as well, rendering his experience visible by publishing poems in *Negro Digest* that caught the attention of artists such as Gwendolyn Brooks and Dudley Randall, who even commissioned a poem from Knight on the anniversary of the assassination of Malcolm X. The poem, "For Malcolm: A Year After," depicts the author's ambivalence about using European structures—"adhere to foot and strict iamb"—to contain his anger and grief over the death of the Afro-centric activist. Knight develops here his sense of form as a survival mechanism that can "control the burst of angry words" before they "boil and overflow/And drench me, drown me, drive me mad" (Knight 1968, p. 43). Yet he also molded and modified form to suit his needs, as when he pairs iambic tetrameter with vernacular by playing with spelling—transforming "rhyme," for instance, into "rime"— while boasting in the collection's biographical notes, "I was aware of what I was doing to the King's English. And that's my biggest bug. I mean—I don't think in proper English, therefore my expressions (and ideas) are not in proper English" (p. 104). These lines point to Knight's insistence that incarcerated writers must alter the physical environment of the poem (stanzas, or little rooms) as well as the physical environment of the prison itself in order to survive. His poem "Cell Song" articulates this sentiment, as well. Here, language holds the potential to transform his solitary cell into a space for reinvention: "I alone . . . twist the space/ with speech," he writes (Knight 1968, p. 11).

Knight's poetry and correspondence also, however, carefully document the ways in which the prison physically alters its residents. In her own encomium for Malcolm X, Knight's partner Sonia Sanchez (1969) articulates survival, or mere "living," as a cheap compromise that pales against Malcolm's revolutionary fervor—"do not speak to me of living./life is obscene with crowds/of white on black./death is my pulse" (p. 39). Her letters to the incarcerated Knight, by contrast, are soft, nurturing, even pleading. Rather

than mobilizing Knight for resistance, Sanchez makes a plea for his survival: "Take care of yr/self—u must survive to tell the world abt itself," she writes in 1968, the same year he is paroled (Collins 2012, p. 7).

The importance and precariousness of survival in a prison setting are major themes within Knight's own poetry from Indiana State Prison, published by Randall's Broadside Press as *Poems from Prison* (1968). In this collection, Knight commemorates those who suffer bodily harm in two vivid character portraits, "For Freckle-Faced Gerald" and "Hard Rock Returns to Prison from the Hospital for the Criminal Insane." Together, these poems depict the wide spectrum of the penitentiary's casualties: Gerald is overtaken because he is too "soft," Hard Rock because he is too "hard."

"For Freckle-Faced Gerald," a poem Knight wrote following a prison rape (Rowell 1996, p. 977), depicts a teenager who is victimized on account of his youth and naiveté: "Sixteen years hadn't even done/a good job on his voice. He didn't even know/how to talk tough or how to hide the glow/of life before he was thrown in as 'pigmeat'/for the buzzards to eat" (Knight 1968, p. 14). Gerald is also "pigmeat" because of his inability to earn their trust and protection. Since Black prisoners are punished for congregating, there is no "safety in number like back on the block"; what is more, "Gerald could never quite win/with his precise speech and innocent grin/ the trust and fists of the young black cats." This short poem, in which the term "buzzards" appears three times, depicts a prison culture in which the vulnerable are quickly cannibalized. Buzzards hover around Gerald, waiting to "light upon his back" in the night, mirrored by the "wiser and bigger buzzards"—prison administrators, policymakers, and other government officials—who have "plotted" the same physical domination and exploitation through structural violence, as Reginald Dwayne Betts (2018) has argued. Knight uses Gerald and the conceit of the buzzards to make visible the sexual abuse and victimization that is a part of the prison landscape. He also uses Gerald to demonstrate the power of the state to create an environment in which smaller predators will perform the work of the "wiser and bigger buzzards."

Knight's portrait of a lobotomized prisoner in "Hard Rock Returns to Prison from the Hospital for the Criminal Insane" offers another look at bodily invasion, though it is experienced this time by a rebel who was "'known not to take no shit/ From nobody,' and he had the scars to prove it" (Knight 1968, p. 11). Released back into general population "like a freshly gelded stallion," Hard Rock now functions as a deterrent and an example of the dangers that accompany resistance in a total institution. Knight also focuses on bodily disfigurement through mixed metaphors, as Hard Rock's lobotomy is figured here as a castration. This analogy explicitly draws on the history of lynching and castration rituals in the U.S. whereby "the mob severs the black male from the masculine" (Wiegman 1993, p. 446); it also conjures the scientific racism from the turn of the century that advocated for castration as an alternative to lynching based on stereotypes about Black masculinity and hypersexuality (Stein 2015, pp. 220–21). The threat Hard Rock presents here is incorrigibility, and his lobotomy suggests that the brain may be the organ that most threatens state authority.

"Hard Rock" also reveals another important and complicated dimension of survival: survivors' guilt. An early version of the poem likens the speaker and his peers, who have gathered round Hard Rock to observe his transformation, to "indians at a corral" (Knight 1968) while a later version likens them to "a herd of sheep" (Knight 1986). These metaphors for entrapment and passivity, respectively, reveal the speaker's recognition that they have all been bearing witness to Hard Rock's rebellions with some degree of hope. Having lived vicariously through his conquests, they are "crushed" by what his defeat symbolizes: "He had been our Destroyer, the doer of things/ We dreamed of doing but could not bring ourselves to do" (Knight 1968, p. 12). Knight's "sheep" metaphor also highlights the speaker's awareness that his own passivity has secured his own survival at the expense of Hard Rock's. Fear is personified in the poem's final lines as a bodily whipping under chattel slavery: "The fears of years, like a biting whip,/Had cut grooves too deeply across our backs" (p. 12). Indeed, fear of further punishment and additional time deters Hard

Rock's peers from behaving like Hard Rock. Survival here is figured through literal and metaphorical scars that serve as a warning to others.

In his attempt to avoid the unsuccessful models that Gerald and Hard Rock represent, Knight turned to writing as a means of survival, often describing his creative process as a struggle to hang onto his lucidity and his humanity. This struggle was exacerbated by the physical isolation imposed by solitary confinement, or "the hole," which also involved, by Knight's account, being stripped naked and immersed in total darkness. While Knight figures poetry as a form of resuscitation, he figures solitary confinement, by contrast, as a kind of death, "the nearest thing I can imagine to being in the grave" (McCullough 1982, p. 3). If bodily incursion represents one extreme of the physical hardships one endures in prison, solitary confinement represents the other. The poem "The Idea of Ancestry," Knight's attempt to place himself within his immediate family, was born out of one such experience:

> I had just gone through a thing of being in the hole for days. The first two or three days in the hole, you sing songs, recite Shakespeare, masturbate, and think about the streets. After about ten days in there, you stop singing songs and start remembering your early life. . . . To keep your sanity, you have you place yourself in the context of the world somehow. I had just been in the hole some thirty or forty days and that poem came. (Rowell 1996, p. 978)

The very composition of the poem is a survival strategy as Knight struggles to maintain his wits during his isolation. Yet Knight also articulates a genealogical and, ultimately, biological sense of survival in the poem itself, which begins with a makeshift family tree of "47 black faces" taped to his cell wall. Here, he documents those who have not survived in parentheses—"grandmothers (1 dead), grand/ fathers (both dead)"—including those who, like the speaker, have been institutionalized: "2 aunts (1 went to the asylum)" (Knight 1968, p. 16). Most disturbing to the family is the "empty space" that represents Knight's uncle who "disappeared" at age 15, and who is "discussed each year/ when the family has a reunion" because he "causes uneasiness in/ the clan." Knight's own incarceration renders him another empty space and source of "uneasiness" within this family lineage, as well.

Knight consistently defined art as a survival technique that provides a brief but profound respite from the suffering and dehumanization of prison life. It is the reversal of the self-alienation that begins with the identification number that is assigned to each resident:

> You're given a number, and then, because of that, begins this process of encap-sulation, cutting off of your feelings. Now that is what works against art, works against it entirely, because art is getting in touch with yourself. When the threat and the pain of life in prison is only relieved under certain conditions, at certain times, art happens. (McCullough 1982, p. 3)

In other works, however, Knight suggests that prison neutralizes art rather than vice versa. In "Apology for Apostasy?" he articulates the atmosphere in prison as not just harmful but "poison[ous]" to the creativity and delicacy that fuels poetry: "Soft songs, like birds, die in poison air/ So my song cannot now be candy" (Knight 1968, p. 30). The tensions generated in these personal and artistic articulations reflect the tensions that lie at the very heart of prison literature: if art and prison perform opposite functions, how are they reconciled in the creative works of incarcerated writers?

Perhaps they are not. Knight grapples with this tension in much of his poetry and continues to struggle under its weight even after his release. Like Malcolm X, he continued to use prison as his "major metaphor" for loneliness and isolation, even telling Charles Rowell in the late 1970s, "in all the real senses I am still in prison" (Anaporte-Easton 1996, p. 945; Rowell 1996, p. 975). Yet unlike Malcolm X, Knight was never able to regain his footing after his release. Despite receiving prestigious grants from the National Endowment for the Arts and the Guggenheim Foundation, Knight suffered from addiction and financial debt for the rest of his life, and his creative output never matched—in terms

of either quality or quantity—the work that he had produced in the Indiana State Prison. The correspondence between Knight and his publisher Dudley Randall, which becomes increasingly strained through the publication of *Belly Songs*, reveal the extent of Knight's writers' block and money troubles: Randall even sends Knight money to pay for his car transmission in the summer of 1975 (Knight 1975). Knight had become so institutionalized, so accustomed to the poison air of the prison, that he found it difficult to adjust to life on the outside: "After living/looking out/ for myself alone for eight years I found it difficult to adjust to a married/family/situation," he wrote of the dissolution of his marriage to Sanchez on 5 April 1970 (Knight 1970).

The writings—and lives—of Malcolm X and Etheridge Knight represent different articulations of and responses to the trope of literacy as survival in carceral contexts in ways that are mutually illuminating. Malcolm X was followed by scores of imprisoned Black autodidactic literary successors including George Jackson and Assata Shakur whose life writing explores similar themes and content. The lyric poetry and life trajectory of Etheridge Knight, however, adds formal as well as theoretical layers to these narratives of resistance and survival, especially given Knight's nuanced exploration of the fears generated by physical captivity, bodily harm, and institutionalization. Unfortunately, not much has changed since the time of Knight's writing, especially given the proliferation of "risks" in the form of structural barriers to reading that are now enshrined in policy and law. Indeed, illiteracy has been reinforced by 21st century legislative and judicial decisions that continue to cede control of educational access to prisons in the name of "security."

## 5. Reading Behind Bars in the 21st Century

According to the most recent statistics from the National Assessment of Adult Literacy (National Center for Education Statistics 2003), 56% of incarcerated adults function at basic or below basic reading levels, scoring well below the Institute of Education Science's standards for "proficiency" in English.[10] In 1994, Congress barred people in prison from receiving Pell Grants, "effectively defunding all college programs in U.S. prisons and sparking broader cuts in all levels of educational programming" (Sweeney 2007, p. 2). Fortunately, this ban was lifted in December 2020, paving the way for federal funding of higher education in prison beginning in 2023. In the meantime, however– as prisons continue to trim their budgets– resources have become increasingly scarce even for the most motivated autodidacts: most prison libraries are makeshift entities that rely on donations, and their size and quality vary wildly from one institution to another. Some do not have libraries at all beyond the minimal legal annals that prisons are required to maintain for incarcerated people who wish to work on their cases.[11] Though the American Library Association adopted a "Prisoners' Right to Read" resolution in 2010, the courts have yet to establish reading as a fundamental "right" to which all people, including those who are incarcerated, are entitled (American Library Association 2014). On the contrary, the Supreme Court in *Beard v. Banks* (2006) upheld a Pennsylvania prison's decision to deny secular periodicals to their "most incorrigible" residents. As Megan Sweeney (2007) argues, this decision effectively permits prisons to use reading deprivation as discipline: "The majority opinion in *Beard v. Banks* constructs reading as a privilege that best serves the interests of the penal system when it is denied to uncooperative prisoners" (p. 779).

As such, even prisons with substantial libraries can and do execute bans on reading material, including materials that are sent to residents from outside prison walls. The U.S. Department of Justice grants wardens complete jurisdiction over whether a publication "is detrimental to the security, discipline, or good order of the institution or if it might facilitate criminal activity" (Bureau of Prisons 2011). Such elastic and subjective criteria have enabled prisons to ban material ranging from prison literature to historical accounts of the Civil Rights movement.[12] Rebecca Ginsburg's Education Justice Project (EJP) faced similar challenges at Danville Correctional Center, which not only banned Frederick Douglass' *Autobiography*, Harriet Beecher Stowe's *Uncle Tom's Cabin*, and W. E. B. Du Bois' *The Souls of Black Folk*, but used this ban to suspend the entire program (Nickeas 2019). Fortunately, the

prison was forced to restore the confiscated books following a successful campaign by EJP and its allies.

Meanwhile, most people in prison must continue to rely on unsympathetic courts to block and reverse capricious book bans. In 2014, activist Robert Saleem Holbrook filed a civil suit against employees of the State Correctional Institution at Coal Township with the help of the Human Rights Coalition (*Holbrook v. Jellen at al.*).[13] He attested that "the prison mailroom supervisor at [SCI Coal Township] reflexively denies all books by Black/Latino authors that provide a radical critique of prisons, as well as all publications that contain articles written by prisoners that critique prisons from an adversarial position" (Holbrook 2012). Though Holbrook and his attorneys settled for $95,000 and won a number of claims on summary judgment, Pennsylvania's DOC continues to control reading through new initiatives that include the creation of a "processing center" that screens and reviews all incoming books (Melamed 2018).

In his recent *New York Times* Op Ed, Christopher Blackwell (2022) pleas for state and federal prisons to "create more explicit book restriction policies that clearly define what constitutes a safety threat" as well as clarifying the appeals process when a book is banned. The author of *Angry White Men: American Masculinity at the End of an Era* sent Blackwell a copy of his book following their correspondence about toxic masculinity, and the book was withheld from Blackwell due to "penological objectives." The rejection notice stipulated that "the content could reasonably be thought to lead or add to tensions between groups specifically in a prison setting." Blackwell successfully navigated an appeals process that released the book to him, but the process was a "bureaucratic maze" that required him to justify the book's value. What is more, the appeals process is idiosyncratic, and specific to Blackwell's objection: it did not "automatically make the book available to others" and would not prevent the mailroom from withholding the same book from others.

Unfortunately, the technology that is being introduced to prisons under the guise of access—such as reading tablets—promise additional rather than fewer barriers to reading access in terms of paywalls. West Virginia prisons have contracted with Global Tel Link to introduce pay-per-minute tablets that have been condemned by PEN America (2019). "Not only is this a predatory policy that will actively disincentivize incarcerated people from reading," said James Tager, PEN's deputy director of Free Expression Research and Policy, "but it rewards the state [financially] for being complicit in these restrictions" ("Pay-Per-Minute"). Worse still, the Prison Policy Initiative reports that many facilities are using tablets as justification for banning the donation of physical books (Finkel and Bertram 2019).

As prisons continue to demonstrate, there are many ways to curb literacy. The writings of Malcolm X and Etheridge Knight challenge those of us on the outside to remain attentive to the material conditions that threaten to blot out reading and writing activity from the artists and intellectuals of our present and future. They compel us to attend not just to the "soft songs, like birds" that hover in the air but also to the "poison air" that threatens their very survival (Knight 1968, p. 30). Most importantly, they help us to understand the ways in which literacy and survival in prison are shaped by race and gender, providing a point of entry for locating, articulating, and addressing the risks and rewards of writing and reading behind bars.

Locating these points of entry can also help us to identify promising points of intervention. In 2020, Reginald Dwayne Betts launched the Freedom Reads project, motivated by his lack of access to books during his own incarceration. The initiative aims to place a curated library of 500 books into 1000 prison facilities across the nation. Recognizing the physical and spatial constraints of the prison, the project also produces handcrafted bookshelves that are tailor made for each facility. The modular system created by the MASS Design Group consists of "bookshelves with intermittent seating, customizable to distinct settings: from a corner of the dayroom to a converted cell" (Betts n.d.). In 2021, Betts received a MacArthur Genius grant to support this project. The first library was installed

in Malcolm X's former cell in Norfolk (Flood 2021). "[With] a project like this," said Betts, "What better place to begin?"

**Funding:** This research received no external funding.

**Institutional Review Board Statement:** Not applicable.

**Informed Consent Statement:** Not applicable.

**Data Availability Statement:** Not applicable.

**Conflicts of Interest:** The author declares no conflict of interest.

## Notes

1    See, for example, the ACLU's material on solitary confinement: https://www.aclu.org/issues/prisoners-rights/solitary-confinement (accessed on 14 December 2022, American Civil Liberties Union n.d.).

2    As early as 1833, ESP's warden noted that nearly one in five residents could not read or write, but the prison did not implement literacy tutoring until the 1840s following Charles Dickens' public excoriation of ESP's system of solitary confinement (Schorb 2014, pp. 122, 127–31). The congregate system did not sincerely invest in education until Eliza Farnham relieved Elam Lynds at Sing Sing in 1844 (p. 169).

3    For a discussion of this era of "corrections" within U.S. penal history, see (Conover 2000, pp. 202–3).

4    Though he was sentenced to eight-to-ten years, X only served seven; he was released early on parole.

5    Recent historical scholarship has questioned whether Malcolm may have exaggerated his lack of literacy in order to render his transformation more dramatic. See Tucker's reading of archival material from Malcolm's prison years, which suggests that Malcolm had an enduring interest in educational opportunities and took advantage of college-level classes at Norfolk.

6    Alas, we don't know much about Malcolm's participation in these courses because he does not describe them in his autobiography. His prison record indicates that he enrolled in two classes– Latin and Great Books, which he did not finish (Tucker 2017, pp. 204–5).

7    The self-curated image of Malcolm as solitary "monk" should be augmented here by recent work on Malcolm's early collective organizing with the Nation of Islam in prison (Felber 2020, p. 38).

8    Following his first stint at Charlestown, Malcolm also served fourteen months at the Massachusetts Reformatory at Concord, where he worked in a furniture shop. While Malcolm does not discuss Concord in his autobiography, his biographers indicate that the facility was, aside from its working toilets, very similar to Charlestown in climate and culture. Here, Malcolm was plagued with health problems most likely brought on by stress (Perry 1991, pp. 112–13; Marable 2011, pp. 74–75).

9    In an important ruling, U.S. Middle District Chief Judge Christopher C. Conner states that the law is "manifestly unconstitutional" and that "the First Amendment does not evanesce at the prison gate" (Miller 2015).

10   *Below Basic* indicates that an adult has no more than the most simple and concrete literacy skills. *Basic* indicates that an adult has the skills necessary to perform simple and everyday literacy activities. *Intermediate* indicates that an adult has the skills necessary to perform moderately challenging literacy activities. *Proficient* indicates that an adult has the skills necessary to perform more complex and challenging literacy activities ("Literacy").

11   The Supreme Court ruled in Bounds v. Smith (1977) that people in prison must have "meaningful access to the courts," though its ruling in Lewis v. Casey (1996) reversed requirements for prison libraries to provide adequate holdings and legal consultation for people in prison.

12   One civil rights organization in Texas has found that these are the two most popular genres of banned books within the Texas prison system ("Banned Books", Texas Civil Rights Project 2011). For scholarship on the arbitrary restrictions placed even on religious writings, see Sullivan.

13   See a summary provided by the Abolitionist Law Center, of which Holbrooks now serves as executive director: https://abolitionistlawcenter.org/our-work/completed-cases/robert-saleem-holbrook/ (accessed on 14 December 2022).

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
