# Peer review of "Reading Behind Bars: Literacy and Survival in U.S. Prison Literature"

_humanities, doi:10.3390/h12010002_

Round 1
Reviewer 1 Report
I enjoyed reading this essay about literacy and prison. It explores a tension between literacy, functioning as a mode of survival, and the harsh conditions of the prison. Drawing on the work of Malcolm X and Etheridge Knight, the essay illustrates this tension nicely. I recommend that the manuscript be accepted with revisions.
The author is clearly in favor of education programs and access to books in prisons, referring to the "unmistakable benefits of literacy." I share this commitment, but I believe the author needs to engage more richly with scholarship exploring the problematic and uncritical faith many have in literacy. See, for example, Harvey Graff's 1970 The Literacy Myth (chapter 6), which analyses the relationship between literacy and crime. Engaging with this work, I believe, will support the author's attention to the specific contexts and material conditions that impact literacy learning. These authors do not deny literacy's potential but urge us to resist blind faith in them.
The author begins by reflecting on his teaching in prison, moving from Golding's text to Donald Goines' street lit and Iceberg Slim with great success only to have restrictions imposed. This anecdote highlights that issues of controlling and banning reading material persist.
On page 2, the author writes, "Building on James' critique, this essay contends that literacy narratives are also responsive to and conditioned by the debasing material conditions of the prison." I agree but view this as an obvious statement; I've never read a narrative about literacy in prison that did not attend to the material conditions and the lived histories of those in prison. I also need to see more clearly how this builds on James's work. Can the author provide some examples of texts on literacy in prison that ignore the specificity? If not, I would eliminate this.
On pages 2-4, the author explores the early history of Auburn and ESP. Yet I wonder why the author does not discuss the history of Pell Grants and the various literacy initiatives that emerged throughout the 1960s and 1970s. (I see that some of this comes in at the end). To provide a comprehensive history of literacy in prison is, of course, beyond the scope of this manuscript. Yet, I still needed clarification as to why we spent so much time with this early history.
The analysis of Malcolm X and Etheridge Knight is excellent and leads nicely to the concluding section on the importance of such reading materials.
I hope this information is helpful, and I look forward to seeing this manuscript in print.
Author Response
Thank you for your excellent feedback! I have provided detailed notes in a cover letter (attached) about how I have incorporated your insights into my revision, along with the feedback from other other two readers. I especially appreciate the way you report encouraged me to nuance my argument about literacy's value. Thank you for your careful reading.
Reviewer 2 Report
I have attached all my comments in a file here.
I think that this is work that should be published, but that it could be improved before publication in ways that would help its own focus and argument, and also help offer a more sizeable contribution to the field.
I have no comments which cannot be shared with the author. I have tried to engage intellectually with the piece, and still to show that this engaging and important work.

Author Response
I can't thank you enough for giving my essay such a thorough and careful reading. I have spent a lot of time revising the piece to address each of your concerns. I try to provide an overview of these revisions in the attached cover letter. I hope that you are also able to view the revised essay to see how I have incorporated each of your revision suggestions through the "track changes" function. In short, your comments really challenged me to nuance and deepen my argument, and I'm especially grateful for the Simon Rolston references, which are now incorporated into the essay, as well. Thank you for this deep engagement with my article, which is much improved as a result!
Reviewer 3 Report
This is excellent. Clearly argued and compelling in linking present struggles with the history of prison reading, and with the examples of Malcolm X and Etheridge Knight. The bridge back to the 21st century from the dives into those 2 prison readers/writers could perhaps be a bit more developed, but that's a minor quibble. I understood how the contexts set up the closing discussion and found the full argument to be a valuable addition to current work on this topic.
Some very minor suggestions:
Footnote 3: Replace "Lynds boasted that his regime delivered “around fifteen hundred lashings were inflicted..." with "Lynds boasted that under his regime “around fifteen hundred lashings were inflicted..."
Consider using "Attica prison uprising" rather than "riot"
Block quotations on p. 11 need 1" indent or other demarcation
Along with the good news for readers inside that is Freedom Reads, you might note that after an activist campaign by EJP and allies IL DOC was forced to restore the confiscated books. And while J-Pay tablets remain a scourge as described, another activist campaign at least stopped them from selling incarcerated people copies of books obtained by J-Pay for free from Project Gutenberg.
Author Response
Thank you for such a careful engagement with my essay! I have made all of your revision suggestions, which I specify in the attached cover letter. I appreciate your attention to detail, especially of language ("uprising" vs. "riot") as well as outcome (great to hear that EJP was able to reverse the book ban). Thank you for your feedback and encouragement on this piece.